# Wheelchair service provision education in Canadian occupational therapy programs

Ed M. Giesbrecht[1]*, Paula W. Rushton[2,3], Evemie Dubé[3]

1 Department of Occupational Therapy, University of Manitoba, Winnipeg, Manitoba, Canada,
2 Occupational Therapy Program, University of Montreal, Montreal, Quebec, Canada, 3 CHU Sainte Justine Research Centre, Montreal, Quebec, Canada

* ed.giesbrecht@umanitoba.ca

**Data Availability Statement:** The minimal data set is included in a Supporting Information file; a small portion of the raw data has been redacted to protect the identity of the survey respondent sites.

## Abstract

Occupational therapists in Canada play a central role in wheelchair service provision. Inadequate entry-to-practice professional education has been identified as a major concern in the delivery of wheelchair related services. The goal of this study was to describe the current education provided in Canadian occupational therapy programs and to map this content against the recommended WHO 8-step wheelchair service provision process. The study used a descriptive cross-sectional online survey design. Educators were recruited from accredited occupational therapy programs in Canada. Participants completed a short socio-demographic questionnaire and a survey with 97 closed- and open-ended questions regarding the wheelchair service provision education provided in their curriculum. Survey data was then mapped according to the WHO 8-step wheelchair service provision process. Twenty-nine educators from all Canadian occupational therapy programs (n = 14) were enrolled. Most participants (55.2%) were full-time faculty members that had been teaching in occupational therapy programs for an average time of 10.9 years. All programs covered at least 4 of the WHO recommended steps, but only 5 programs covered all steps. *Assessment* and *Prescription* steps were covered in every program while the *Referral & Appointment*, *Funding & Ordering*, *Fitting* and *User Training* steps were covered in most programs. The pedagogic approach, the amount of time dedicated to wheelchair-related content, and the type of evaluation used varied greatly between programs. This study is the first to provide a detailed description of wheelchair service provision education across all Canadian occupational therapy programs according to the WHO 8 steps and provides a foundation for collaborative efforts to promote best practice in entry-to-practice professional education.

## Introduction

Personal mobility is a fundamental human right, according to Article 20 of the United Nations Convention on the Rights of Persons with Disabilities [1]. For those with a mobility impairment, a wheelchair is an essential assistive technology. Wheelchair provision is a multifaceted and complex intervention, requiring the provider to consider the interaction of the person (physical, cognitive, affective components), environment (physical, social, institutional)

**Funding:** PWR and EMG received funds for this work from the Craig H. Neilsen Foundation (https://chnfoundation.org/, Grant #578562).The funders had no role in study design, data collection and analysis, decision to publish, or preparation of the manuscript.

**Competing interests:** The authors have declared that no competing interests exist.

and activities of choice (self-care, productivity and leisure) [2]. Appropriate wheelchair service provision addresses the barriers that compromise an individual's functioning [2], independence [3] and well-being [2]. Inappropriate wheelchair service provision can result in poorer health outcomes (e.g., physiological complications [4] and pressure injuries [5]), decreased functional abilities [6], decreased quality of life [7], social isolation [8], exclusion [8], and even death [8].

To promote best practice, the World Health Organization (WHO) published wheelchair provision guidelines in 2008 [9], in which they described an eight-step process: *Referral & Appointment*; *Assessment*; *Prescription*; *Funding & Ordering*; *Product Preparation*; *Fitting*; *User Training*; and *Follow-up, Maintenance & Repairs*. Evidence from subsequent studies implementing this process report positive impacts on wheelchair user satisfaction, participation, health, quality of life, daily wheelchair use, and activities of daily living [10–14]. The health care professionals primarily responsible for wheelchair service provision (i.e., occupational therapy [OT], physical therapy [PT], and prosthetics & orthotics [P&O]) often lack the competencies required for comprehensive service delivery [14]. The need for a competent workforce is a pressing issue for countries around the world as evidenced by prioritization in the WHO Global Cooperation on Assistive Technology [15], the 2017 Global Priority Research Agenda [16], the first Global Research, Innovation, and Education on Assistive Technology Summit [17, 18], and the 2018 Wheelchair Stakeholders' Meeting [19]. A recent position paper posits that capacity building and delivery of adequate education and training for all wheelchair service providers is critical to developing sustainable wheelchair provision systems [20].

Entry-to-practice education programs are pivotal in ensuring clinical competence. Emergent evidence benchmarking preparatory wheelchair service provision training against the WHO 8-step process [9] highlights shortcomings in current educational programs. Globally, ~ 21% of OT, PT and P&O university programs do not include any wheelchair provision education and, among those that do, there is considerable variability in content, pedagogical approach, evaluation and hours taught [21]. In low- and lower-middle income countries, ~75% of programs lack any content on wheelchair provision [14]. In Colombian occupational therapy (n = 7) [22] and physical therapy (n = 2) programs [23], student (n = 199) knowledge does not align with the WHO 8-step wheelchair service provision process. Programs in Romania [24] and the Philippines [25] have identified a need to improve wheelchair service provision in their rehabilitation programs. A recent World Federation of Occupational Therapists survey (n = 1050) confirmed a lack of professional preparation with 29% (n = 305) of respondents indicating they received insufficient or no training on wheelchair provision [26]. This evidence highlights insufficient entry-to-practice training among providers of wheelchair service [14, 19].

Most OT, PT and P&O university programs are approved according to international minimum education standards and/or accredited by national organizations. For example, all but 357 of the 96,551 OT programs worldwide are approved by the World Federation of Occupational Therapists (WFOT) [27]. In Canada, all 14 OT entry-to-practice programs are approved by WFOT [27] and accredited by the Canadian Association of Occupational Therapists (CAOT) [28]. However, while each programs' curriculum is WFOT approved and CAOT accredited, the incorporated content is not prescribed. As a result, there is variability across programs including wheelchair service provision specific content. Existing literature on entry-to-practice wheelchair provision education in Canada is sparse, but there is evidence that education is lacking in the *User Training* step [29] and this lack of preparation is reflected in limited implementation in clinical practice [30–32]. Surveying and documenting wheelchair service provision education delivered in Canadian OT programs would provide a more accurate and in-depth understanding of national training practices. Furthermore, this could

highlight gaps in education relative to the WHO 8-step process, identify where the greatest diversity exists among programs, and provide insights into potential opportunities for sharing of resources and strategies. Thus, the objective of this study was to describe wheelchair service provision education offered in Canadian OT entry-to-practice programs and map the content to the WHO 8-step wheelchair service provision process. This study is a first step towards the development of a national strategic plan to enhance wheelchair service provision education and develop a Community of Practice among Canadian OT program educators in this area. This vision has the potential to impact over 1000 students who graduate from these programs each year [28] and further the development of a competent workforce of wheelchair service providers in Canada.

## Materials & methods

### Design

This study employed a descriptive cross-sectional online survey design. It is part of a larger study aiming to develop a profile of wheelchair education provided in Canadian university occupational therapy curricula and a strategic plan for addressing identified gaps. The study was approved by the Research Ethics Board of the CHU Sainte-Justine (#2020–2336).

### Recruitment

A volunteer sample was recruited from the 14 accredited Canadian OT programs. An invitation was sent via email to the director of each program, requesting 1–3 educators to participate. Individuals were eligible to participate if they were (1) a faculty member or guest lecturer in the OT program; (2) currently teaching wheelchair content or were in the process of planning to teach wheelchair content; and (3) able to read and speak in English or French (the two official Canadian languages). Recruitment took longer than expected due to the onset of the COVID-19 pandemic; the online survey remained open from August 2019 to May 2021. Participants provided written informed consent prior to participation and were not reimbursed for their time.

### Measurement

The research team, composed of two wheelchair service provision education experts, developed the online survey, which was designed to describe the current wheelchair service provision education offered in the OT program in relation to the WHO 8-step process. The initial iteration of the survey was piloted by four graduate students and one research assistant who provided feedback on the clarity of questions and an estimated completion time of 30–40 minutes. The final version consisted of 97 closed and open-ended questions. The survey was divided into three sections: (1) university program demographic and descriptive information (5 questions); (2) identification of the educators completing the survey (2 questions); and (3) wheelchair service provision content, instructional method and evaluation according to the WHO 8-steps of wheelchair provision (90 questions). If a respondent indicated they did not include content on a specific WHO step, the survey would skip over the follow-up questions and immediately move to the next WHO step. Close-ended questions were either dichotomous (yes/no) or multiple-choice. Open-ended questions were to specify the teaching resources used, the "other" category or to provide the number of courses or hours dedicated to wheelchair-related content. In addition, each educator completed a 16-item sociodemographic questionnaire about their own personal information (e.g., age, sex, education, faculty position,

teaching experience). Both the survey and sociodemographic questionnaire were administered using SurveyMonkey (SurveyMonkey Canada Inc., Ottawa, Canada).

## Data collection

Upon receiving informed consent to participate in the larger study, educators were emailed the link to the short sociodemographic questionnaire. Once this questionnaire was completed by each educator recruited from the OT program, the link to the survey was emailed with the request that one survey per program be collaboratively completed. The Tailored Design Method [33] was used to maximize response rate, in that reminder emails were sent to programs who had not yet completed the survey 2, 4 and 6 weeks following the initial email invitation.

## Data analysis

Survey data were exported from SurveyMonkey into SPSS Statistics (IBM Corp, Armonk, New York) Version 26.0 for analysis. Data were analyzed using descriptive statistics (means, standard deviations, frequencies). The teaching resources provided in the open-ended survey questions and shared with the researchers were categorized as in-house content (e.g., power point slide decks, case studies, lab guides, online modules), provincial/local resources (e.g., local practice guidelines, provincial forms, supplier forms, rehabilitation centers forms, display of products by vendors) or open-source resources (e.g., training programs, textbooks, articles). The research team then conducted a systematic mapping process to compare the education offered in each program compared to the WHO 8-steps to identify patterns in education.

## Results

### Participants

A total of 29 educators were enrolled, representing all 14 Canadian OT programs (average of 1.86 per program, median 1, range 1 to 5). All 14 entry-to-practice programs were master's level; the 5 programs in the province of Quebec were Bachelors-Masters continuum programs. Participants reported that programs ranged in total duration from 2 to 5 years and that 9 of 14 (64%) were taught in English. The average number of students admitted per year was 78 (range 38–140). Sociodemographic characteristics of the study participants are described in Table 1. Participants ranged in age from 31 to 63 years (48.3 years ± 7.9) and were mostly women with full-time faculty member positions. Most participants (79.3%) had taken additional courses specific to wheelchair service provision since the completion of their clinical degree, but only 4 participants had additional certification related to wheelchair service provision (e.g., from the International Society of Wheelchair Professionals ISWP).

### Description of wheelchair-related content

Only 5 programs covered all the WHO recommended 8 steps of wheelchair service provision, with all programs covering at least four of the eight steps. Step 2 (*Assessment*) and 3 (*Prescription*) were covered in all programs. Most programs also covered step 1 (R*eferral & Appointment*), 4 (*Funding & Ordering*), 6 (*Fitting*) and 7 (*User Training*). Step 5 (*Product Preparation*) and 8 (*Follow-up, Maintenance & Repairs*) were the least frequently covered steps. The approximate amount of time dedicated to wheelchair-related content varied greatly between programs from 2.5 to 48 hours (Table 2).

**Table 1. Participants' sociodemographic characteristics.**

| Characteristic | Value |
|---|---|
| Age in years, mean (SD, range) | 48.3 (7.9, 31–63) |
| Gender, n (%) | |
| Man | 3 (10.3) |
| Woman | 25 (86.2) |
| Non-binary | 1 (3.4) |
| Primary Language, n (%) | |
| French | 7 (24.1) |
| English | 22 (75.9) |
| Highest Level of Education, n (%) | |
| Bachelor | 6 (20.7) |
| Master (OT/research) | 11 (37.9) |
| PhD | 12 (41.4) |
| Years teaching in OT program, mean (SD, range) | 10.9 (6.9, 1–29) |
| Position in OT university program, n (%) | |
| Full-time faculty member | 16 (55.2) |
| Part-time faculty member | 1 (3.4) |
| Adjunct faculty member | 5 (17.2) |
| Invited presenter | 2 (6.9) |
| Clinician | 4 (13.8) |
| Clinical faculty | 1 (3.4) |
| Currently practicing clinically, n (%) | 15 (51.7) |
| Practice includes wheelchair service provision, n (%) | 12 (41.4) |
| Years teaching wheelchair content, mean (SD, range) | 9.2 (7.2, 0–30) |
| Role in delivery of wheelchair education, n (%) | |
| Coordinate a course that is exclusively wheelchair-specific | 3 (10.3) |
| Coordinate a course that includes wheelchair-specific content | 13 (44.8) |
| Teach within a course that is exclusively wheelchair-specific | 3 (10.3) |
| Teach wheelchair-specific content within a course | 14 (48.3) |
| Assist within a course that is exclusively wheelchair-specific | 2 (6.9) |
| Assist with wheelchair-specific content within a course | 6 (20.7) |
| Other wheelchair-relevant involvement | 5 (17.2) |
| Participated in developing wheelchair course, n (%) | 23 (79.3) |
| Continuing education obtained on wheelchair service provision, n (%) | 23 (79.3) |
| Continuing Education offered through a university program | 2 (6.9) |
| Conference workshop | 21 (72.4) |
| Educational opportunity offered by a private organization | 8 (27.6) |
| Online course | 2 (6.9) |
| Self-study | 15 (51.7) |
| Other | 4 (13.8) |
| Certification related to wheelchair service provision, n (%) | 4 (13.8) |
| Member of one or more wheelchair-related organizations, n (%) | 10 (34.5) |
| Rehabilitation Engineering and Assistive Technology Society of North America | 9 (31.0) |
| International Society of Wheelchair Professionals | 7 (24.1) |
| International Society of Prosthetics and Orthotics | 2 (6.9) |
| Other | 4 (13.8) |

**Table 2. Time allotted to teaching each of the WHO wheelchair service provision steps by program.**

| Program | Step 1 Referral & Appointment | Step 2 Assessment | Step 3 Prescription | Step 4 Funding & Ordering | Step 5 Product Preparation | Step 6 Fitting | Step 7 User Training | Step 8 Follow-up, Maintenance Repairs | Total time | Number of steps covered |
|---|---|---|---|---|---|---|---|---|---|---|
| 1 | | 18h | 9h | 4h | 4h | 4h | 6h | 3h | 48h | 7 |
| 2 | 0.17h | 2.3h | 0.5h | 0.17h | 2.3h | 0.5h | 0.33h | 1.5h | 7.8h | 8 |
| 3 | 2h | 7.5h | 6.5h | 2h | 3.5h | 1h | 6h | 1h | 29.5h | 8 |
| 4 | 0.25h | 2.5h | 3h | 0.25h | 0.25h | 0.5h | 0.25h | 0.25h | 7.3h | 8 |
| 5 | 0.25h | 1h | 0.5h | 0.5h | | | 0.25h | | 2.5h | 5 |
| 6 | 1h | 6h | 4h | 1h | | 3h | | 1h | 16h | 6 |
| 7 | 1h | 21h | 10h | 2h | 0.33h | 0.75h | 13h | 0.67h | 48.8h | 8 |
| 8 | 1h | 5h | 5h | 1h | | 5h | | | 17h | 5 |
| 9 | | 3h | 2h | | | 2h | 1h | | 8h | 4 |
| 10 | 1h | 9h | 6h | 1h | | 2h | | | 19h | 5 |
| 11 | | 4h | 1h | 1h | | | 2h | | 8h | 4 |
| 12 | 0.5h | 6h | 2h | 0.5h | 0.5h | 0.5h | 1.5h | 0.5h | 12h | 8 |
| 13 | 2h | 8h | 10h | 2h | 3h | 2h | 2h | | 29h | 7 |
| 14 | 0.17h | 3h | 3h | 0.5h | | 3h | 2h | | 11.5h | 6 |
| Mean | 0.7h | 7h | 4.5h | 1.1h | 0.98h | 1.7h | 2.5h | 0.57h | 18.9h | |
| Median | 0.5h | 5.5h | 3.5h | 1h | 0.13h | 1.5h | 1.3h | 0.13h | 14h | |
| Standard deviation | 0.7h | 5.9h | 3.4h | 1h | 1.5h | 1.6h | 3.6h | 0.85h | 14.8h | |
| Number of programs | 11 | 14 | 14 | 13 | 7 | 12 | 11 | 7 | | |

## Delivery of wheelchair-related content (pedagogic approach, resources, evaluation)

The integration of wheelchair service provision education in the curricula of participating programs is described in Table 3. For most programs, wheelchair related content was distributed throughout the curricula as part of mandatory courses. Only a limited number of programs offered wheelchair related content in full mandatory courses or full optional courses. Lectures and laboratories with instructors were the most frequent pedagogic approaches used to deliver wheelchair content. Educators were mostly using written evaluation to evaluate students' knowledge in each step with step 4 and 8 being the less evaluated. Other methods of evaluation included practical evaluation and oral presentations. Two programs (14.3%) also reported that data about students' learning was collected outside of course-related evaluation using the ISWP Basic Test [34] and administering wheelchair-related outcome measures before and after some components (i.e., skills training). Most programs (93%) used in-house content combined with open-source resources and provincial/local resources. Provincial/local resources were mostly forms or guides from provincial governments, local rehabilitation centers and wheelchair suppliers. Open-source resources that were the most used by educators were documents from the Rehabilitation Engineering and Assistive Technology Society of North America [35], the World Health Organization [36, 37] and the Wheelchair Skills Program [38].

## Discussion

This study was novel in several regards. First, this study is the first of which we are aware that explored specific practice in education and evaluation of wheelchair service provision content

Table 3. Delivery, teaching and evaluation approaches for each WHO wheelchair service provision step.

| WHO Step | Programs included | Course Format | | | | | | | Teaching Method | | | | | | Evaluation | | | | |
|---|---|---|---|---|---|---|---|---|---|---|---|---|---|---|---|---|---|---|---|
| | | Only part of a course | Only full courses | | | Part of a course and Full mandatory course | Part of a course and Full optional course | Full courses (mandatory and optional) and part of a course | Lecture | Lab (I) | Lab (S) | Lab (W) | Online module | Other | Written | Practical | Online | Other | None |
| | | | Mandatory | Optional | Mandatory and optional | | | | | | | | | | | | | | |
| 1 | 11 | 6 (55%) | 0 (0%) | 1 (9%) | 0 (0%) | 4 (36%) | 0 (0%) | 0 (0%) | 10 (91%) | 6 (55%) | 0 (0%) | 3 (27%) | 2 (18%) | 2[a,c] (18%) | 6 (55%) | 3 (27%) | 1 (9%) | 3[e,f] (27%) | 3 (27%) |
| 2 | 14 | 8 (57%) | 0 (0%) | 0 (0%) | 0 (0%) | 4 (29%) | 2 (14%) | 0 (0%) | 10 (71%) | 9 (64%) | 4 (29%) | 5 (36%) | 4 (29%) | 1[c] (7%) | 12 (86%) | 7 (50%) | 2 (14%) | 2[g] (14%) | 1 (7%) |
| 3 | 14 | 7 (50%) | 0 (0%) | 0 (0%) | 0 (0%) | 4 (29%) | 2 (14%) | 1 (7%) | 11 (79%) | 11 (79%) | 3 (21%) | 4 (29%) | 2 (14%) | 2[b,c] (14%) | 11 (79%) | 4 (29%) | 2 (14%) | 2[e,g] (14%) | 1 (7%) |
| 4 | 13 | 5 (38%) | 0 (0%) | 0 (0%) | 0 (0%) | 4 (31%) | 3 (23%) | 1 (8%) | 11 (85%) | 5 (38%) | 0 (0%) | 1 (8%) | 1 (8%) | 0 (0%) | 5 (38%) | 0 (0%) | 1 (8%) | 2[e] (15%) | 7 (54%) |
| 5 | 7 | 5 (71%) | 0 (0%) | 0 (0%) | 1 (14%) | 1 (14%) | 0 (0%) | 0 (0%) | 7 (100%) | 5 (71%) | 1 (14%) | 1 (14%) | 0 (0%) | 0 (0%) | 4 (57%) | 1 (14%) | 2 (29%) | 0 (0%) | 2 (29%) |
| 6 | 12 | 7 (58%) | 1 (8%) | 1 (8%) | 0 (0%) | 3 (25%) | 0 (0%) | 0 (0%) | 11 (92%) | 6 (50%) | 3 (25%) | 1 (8%) | 0 (0%) | 1[c] (8%) | 6 (50%) | 1 (8%) | 1 (8%) | 0 (0%) | 6 (50%) |
| 7 | 11 | 9 (82%) | 0 (0%) | 1 (9%) | 0 (0%) | 1 (9%) | 0 (0%) | 0 (0%) | 8 (73%) | 6 (55%) | 1 (9%) | 1 (9%) | 2 (18%) | 1[d] (9%) | 8 (73%) | 3 (27%) | 2 (18%) | 0 (0%) | 2 (18%) |
| 8 | 7 | 4 (57%) | 0 (0%) | 1 (14%) | 0 (0%) | 2 (29%) | 0 (0%) | 0 (0%) | 5 (71%) | 3 (43%) | 1 (14%) | 0 (0%) | 1 (14%) | 0 (0%) | 2 (29%) | 0 (0%) | 1 (14%) | 0 (0%) | 4 (57%) |

[a] vendors and a wound care day of 5 hours

[b] case-studies with pairs

[c] site visits to local clinics that specialize in wheelchair assessment or prescription

[d] online videos from Wheelchair Skills Program

[e] oral presentations

[f] complete a form for a client

[g] observation during lab and feedback.

Note: Lab (I) = with instructors; Lab (S) = with simulated clients; Lab (W) = with wheelchair users.

across the 8-step framework specific to occupational therapy programs. Second, this study engaged every accredited entry-to-practice university program in Canada, providing a comprehensive national overview. The participation of all 14 programs reflects the programs' prioritization of this content area and commitment to enhancing preparation of occupational therapists for clinical practice.

The total number of individuals involved in teaching wheelchair-specific content is unknown, but our experience suggests that the 29 study participants represent a majority of educators in OT programs in Canada. The composition of study participants reflected a broad spectrum of educator roles, with roughly half being full-time faculty and 40% actively practicing in clinical wheelchair service delivery. They typically had extensive experience coordinating or teaching this content area and nearly 80% had engaged in wheelchair-related continuing education activities. While a third of participants were active members of an assistive technology organization, less than 15% had wheelchair service provision certification.

There was considerable diversity in the time allocated to wheelchair-specific content across programs, ranging from 2.5 to over 48 hours. Likewise, there was a broad range of steps covered across the 14 programs. While half of the programs were quite comprehensive, covering 7 or 8 steps, nearly one third addressed only 4–5 steps. This variability may reflect the lack of wheelchair-specific content requirements in Canadian OT program accreditation standards [20, 21]. Similar issues and patterns of diversity have been reported in other middle- [14] and high-income countries as well [21]. Decisions about which steps to include may relate to relevance identified by course coordinators, perceptions of whether particular steps fall within a regional scope of practice, or prioritization within given time constraints. A post-hoc exploration found a weak correlation ($r = 0.41$) between the amount of time allocated and the number of steps covered, but this was not a statistically significant relationship. The programs appear to fall into three broad groupings with 35% providing 8 hours or less, 35% providing 12–19 hours and 30% providing 26–49 hours. Similarly, there appear to be some regional trends with western programs ($n = 3$; $m = 28.4$ hours covering 7.7 steps) higher than eastern programs ($n = 6$; $m = 18.8$ hours covering 5.7 steps) followed by central programs ($n = 5$; $m = 12.7$ hours covering 6.4 steps), although the small number of programs precludes a meaningful statistical analysis. Individual programs clearly address this content area differently in terms of time explicitly dedicated to wheelchair service provision. Despite evidence suggesting that occupational therapists are more likely to provide [39], and be knowledgeable in, wheelchair service provision than other professions [40], there is clearly room for improvement and greater uniformity in their professional education as suggested by D'Innocenzo [41]. While national accreditation processes require programs to ensure competency across the professional roles defined in the Profile of Practice [42], they do not explicate content-specific requirements [43].

Regarding individual steps, *Assessment* and *Prescription* were universally addressed, suggesting these are high priority topics. In most programs, *Assessment* is given the most teaching time. This is not surprising given the magnitude of resources available and broadly accepted practice standards. Likewise, the *Prescription* step addresses clinical reasoning and application of information garnered during assessment, which is essential to clinical practice. It is influenced to some degree by product knowledge, which is an evolving market and often addressed collaboratively with vendors, which might explain the lower number of hours allocated in most programs. *User Training* was assigned the third highest duration among the 11 programs that included this step. This is a somewhat higher ratio than a previous survey of 11 Canadian OT programs that found 7 included wheelchair skills training, 57% providing less than 5 hours of content [29]. A recent survey among OTs practicing in one Canadian province reported one third felt their professional education inadequately prepared them to provide training to clients and caregivers [31] while a qualitative study among rehabilitation clinicians in the

American Midwest reported very limited education on such training was provided before entering practice [44]. Over 30% of wheelchair service providers in a global survey felt unprepared to deliver wheelchair skills training and 24% reported inadequate professional training as a barrier to offering this service component [39]. Collectively, this suggests the *User Training* step merits further attention in entry-to-practice programs. Wheelchair skills training has a growing body of evidential literature, and typically entails 2–4 hours of experiential learning, using a bootcamp approach, to adequately prepare students for clinical application [45–48]. In our study, programs with fewer total hours tended to spend one hour or less on the *User Training* step; however, most programs that did not address the *User Training* step at all had 16 hours or more of content suggesting this decision was not entirely related to time. The *Funding & Ordering* step was addressed in nearly every program but received fairly brief exposure, approximately one hour on average. Content on this topic typically reflects funding agencies, programs and vendors specific to regional jurisdictions. While Canada has a nationally funded health care system, each province and territory administers its own health insurance plan with vastly differing access to wheelchair products and funding [49]. *Product Preparation* and *Follow-up*, *Maintenance & Repairs* steps were reported by only half of programs nationally. In clinical practice, occupational therapists may have access to technicians, either in-house or through a suppling vendor, whom they rely on to perform many of these functions during wheelchair service delivery. Consequently, educators (many who currently practice in this area) may pragmatically consider this an area of lower priority. A recent analysis of knowledge test performance among wheelchair service providers globally [40] found *Follow-up* to be among the lowest scoring domains, indicating this gap is not unique to occupational therapy and may be an area of priority for curriculum reform. It should be noted that the *Referral & Appointment* step was allocated the least amount of time but was covered in nearly 80% of programs.

The delivery of content across steps was generally spread throughout the duration of entry-to-practice programs. The different aspects of wheelchair service delivery are included both earlier and later in occupational therapy curricula and, in many cases, throughout. A relatively small number of programs offer courses that deal exclusively with wheelchair service delivery; rather, most integrate this information into a broader course or courses within the program. All programs include wheelchair-specific content as part of their core curriculum. Only a small proportion of programs offer elective courses addressing this specific topic; this is not surprising given that Canadian entry-to-practice programs have little, if any, elective content. Lecture was the most prevalent method of content delivery across the 8 steps. Hands-on labs of some sort were used at least 50% of the time, aside from the *Funding* and *Follow-up* steps. Use of authentic and simulated wheelchair users was more common for teaching the *Assessment* and *Prescription* steps, allowing students to observe or practice these skills either on campus or in a clinic setting. This type of experiential learning is optimal, but the lower frequency of use reflects the resource and pragmatic challenges of bringing in clients or volunteers and providing sufficient time/access for students to practice, particularly with larger class cohorts. Only six programs utilized on-line modules, primarily for the *Assessment* step. There appears to be considerable opportunity yet for programs to leverage on-line and asynchronous learning strategies to supplement or support the lecture and lab components of wheelchair-related content, as has been proposed in several publications [20, 22]. It is possible that programs may have begun developing these types of resources during the COVID pandemic [50, 51],which occurred subsequent to most of the survey responses.

With respect to evaluation, written exams were used most commonly across all steps. Practical exams were used less frequently; half of the programs evaluated the Assessment step in this manner. Student evaluations did not necessarily incorporate every step in the wheelchair

provision process even if it was taught. The *Funding*, *Fitting*, and *Follow-up* steps were not evaluated at least half of the time. Given that wheelchair-related content is typically taught as a component in a larger (typically skills-based) course, educators need to constrain and prioritize content that is most essential for evaluation. *Assessment* and *Prescription* were rarely excluded, corresponding with the prevalence of inclusion and time allocated. Likewise, *User Training* was evaluated by most of the 10 programs that included this component.

Several factors should be acknowledged as limitations with this study. While we had respondents from every Canadian OT program, there were typically 1–3 individuals from each site collectively responding to the survey and we might have a more comprehensive summary if we were able to hear from all wheelchair content educators. Two of the study authors have primary responsibility for wheelchair service provision education in their respective occupational therapy programs. To minimize bias in data collection, we enrolled other educators involved in wheelchair-related content delivery from these two programs. While the program-specific responses accurately reflect content and delivery, the absence of the two study authors as participants may have impacted both the demographic profile and level of detail for course content, delivery and evaluation for these two sites. Time allocated to the 8 steps were based on participant estimates and their interpretation of the step descriptions; where participants provided a time "range" we reported the more conservative (lower) value. While data from this study is the most comprehensive to date in terms of Canadian education, it may not be reflective of practice in other national programs or disciplines, particularly those in less-resourced settings. This study provides important quantitative information about how wheelchair-related content is incorporated into Canadian OT programs. Future study could also investigate the quality of this education provided, including student perspectives.

## Conclusions

This study is the first to provide a detailed description of wheelchair service provision education across all Canadian OT programs. It highlights the differing amounts of time allocated to wheelchair service provision education among programs, exposes that only 35.7% of programs cover all of the 8 steps recommended by the WHO and identifies the lack of experiential and online pedagogic strategies used for both content delivery and evaluation. These findings provide a foundation for subsequent collaborative efforts to promote best practice in entry-to-practice professional education at a national level. Future research should investigate how programs negotiate extent and scope of wheelchair service provision content, including barriers and facilitators of change and innovation.

## Supporting information

**S1 File. Sociodemographic questionnaire.**
(PDF)

**S2 File. Educator survey.**
(PDF)

**S3 File. Redacted.**
(XLSX)

## Acknowledgments

We thank Karen H. Fung for her contributions during recruitment and preliminary data collection and analysis.

## Author Contributions

**Conceptualization:** Ed M. Giesbrecht, Paula W. Rushton.

**Data curation:** Paula W. Rushton.

**Formal analysis:** Evemie Dubé.

**Funding acquisition:** Ed M. Giesbrecht, Paula W. Rushton.

**Investigation:** Ed M. Giesbrecht, Paula W. Rushton.

**Methodology:** Ed M. Giesbrecht, Paula W. Rushton.

**Project administration:** Ed M. Giesbrecht, Paula W. Rushton.

**Resources:** Paula W. Rushton.

**Software:** Paula W. Rushton.

**Supervision:** Ed M. Giesbrecht, Paula W. Rushton.

**Validation:** Ed M. Giesbrecht, Paula W. Rushton.

**Writing – original draft:** Ed M. Giesbrecht, Paula W. Rushton, Evemie Dubé.

**Writing – review & editing:** Ed M. Giesbrecht, Paula W. Rushton, Evemie Dubé.

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
