## [Decision Letter · Decision Letter 0]

28 Sep 2021

PONE-D-21-27046Wheelchair service provision education in Canadian occupational therapy programsPLOS ONE

Dear Dr. Giesbrecht,

Thank you for submitting your manuscript to PLOS ONE. After careful consideration, we feel that it has merit but does not fully meet PLOS ONE’s publication criteria as it currently stands. Therefore, we invite you to submit a revised version of the manuscript that addresses the points raised during the review process.

The paper needs major revisions.

We look forward to receiving your revised manuscript.

Kind regards,

Alessandro de Sire, M.D.

Academic Editor

PLOS ONE

Journal Requirements:

Reviewers' comments:

Reviewer's Responses to Questions

**Comments to the Author**

1. Is the manuscript technically sound, and do the data support the conclusions?

Reviewer #1: Yes

Reviewer #2: Partly

2. Has the statistical analysis been performed appropriately and rigorously? 

Reviewer #1: Yes

Reviewer #2: Yes

3. Have the authors made all data underlying the findings in their manuscript fully available?

Reviewer #1: Yes

Reviewer #2: Yes

4. Is the manuscript presented in an intelligible fashion and written in standard English?

Reviewer #1: Yes

Reviewer #2: Yes

5. Review Comments to the Author

Reviewer #1: 1) In the abstract in line 28 I advise not to specify the 8 phases because there is no correspondence with the introduction. It can be confusing

2) Materials and methods: Measurement: explain better from line 131 to line 133 (it is not clear)

3) Materials and methods: Measurement: Line 136-139 should belong to section 1, you could insert it before

4) In the materials and methods you could insert the questionnaire to explain better

5) In discussion: from line 220 all numerical evaluations should be included in the results

6) Bibliography should be included in the discussion to support the considerations

Reviewer #2: This cross-sectional study aims to picture the current education in occupational therapy programs in Canada about the wheelchair service provision process, comparing them to the WHO standards.

This study seems to be engaging in particular for the readers who work in a Canadian's similar setting about the wheelchair prescription. However, it should be noted that the involvement of occupational therapists both in prescription and evaluation of wheelchairs is not a worldwide care practice.

The following major concerns about this study should be considered to guide in the amendment of the article or in the adjunction of the limitations' section, where appropriate:

- the small number of educators involved in comparison to the programs (less than 2 for each occupational therapy program);

- the lack of the student perception about the theme investigated, both on the quality of the programs and the time dedicated to it. Evaluate the quality of the program only investigating the educators' side appears a partial overview;

- the conclusion section does not draw conclusive consideration about the data obtained but repeats the discussion and results.

- the discussion and conclusions on the external validity of the results appears lacking.

Minor issues are listed below:

- Title: the study design is missing.

- Abstract: It is preferable to not use abbreviations in the abstract, except WHO that is worldwide known to all readers.

Methods: the time necessary to complete the survey could be interesting to evaluate the educators' small number of complete answers.

- Results: 29 educators could appear as a high number if the total number of possible educators engageable is not stated.

- Discussion: It seems unnecessary to report both in minutes and hours the time allocated to the programs; choose one of them.

6. PLOS authors have the option to publish the peer review history of their article (what does this mean?). If published, this will include your full peer review and any attached files.

Reviewer #1: **Yes: **Dalila Scaturro

Reviewer #2: No

---

## [Author Response · Author response to Decision Letter 0]

8 Oct 2021

Response to Reviewers

Dear Reviewers:

Thank you for the constructive and comprehensive feedback on our manuscript. We have outlined the feedback comments individually below, along with our response and/or revision for each item.

R1. In the abstract in line 28 I advise not to specify the 8 phases because there is no correspondence with the introduction. 

Thank you for this. We have removed “8-steps” from this sentence (line 28).

R1. Explain better from line 131 to line 133 (it is not clear) 

This sentence was revised to improve comprehension (lines 132-134).

R1. Line 136-139 should belong to section 1, you could insert it before 

The questionnaire was separate from the University sociodemographic information, as it related to the respondents individually. We have revised this sentence to make it clearer (lines 137-138).

R1. In the materials and methods you could insert the questionnaire to explain better 

Thank you for this suggestion. We have now included both the Sociodemographic Questionnaire and Educator Survey as Supporting Information with this revision (S1 and S2 Files).

R1. In discussion: from line 220 all numerical evaluations should be included in the results 

The numerical values included here are all drawn directly from Table 2 (Total time column; Number of steps covered column).

R1. Bibliography should be included in the discussion to support the considerations 

Thank you for this suggestion. In addition to the existing citations in paragraph 3 (2 references), paragraph 4 (6 references), and paragraph 5 (2 references), we have further incorporated the literature in support of our findings on lines 228-231, 255-257, and 293.

R2. - the small number of educators involved in comparison to the programs (less than 2 for each occupational therapy program) 

Thank you for this observation. In recruiting for this study, we solicited individuals who taught this specific content area (via recommendation from each program director). In Canadian OT programs, wheelchair-related content is typically taught by a select number of faculty (or clinicians with this expertise are brought in); consequently, the total number of potential respondents aware of wheelchair-specific curricula is small. In our survey we inquired how many individuals contributed to development or teaching of this content and the total across programs was 67, so our participation was 43% of all individuals related in any way to content delivery, which we would consider to be substantial. We have added an acknowledgement of this sub-set in the limitations section (lines 305-308). 

R2. the lack of the student perception about the theme investigated, both on the quality of the programs and the time dedicated to it. Evaluate the quality of the program only investigating the educators' side appears a partial overview 

This is an interesting point. Obtaining student perspectives would provide another insight into wheelchair-related education. We acknowledge our intent was to obtain a quantitative description of content provision (related to the WHO 8-step structure) as an important first step, rather than assess the quality of education – which would indeed also be an important and interesting study. We have taken your suggestion and incorporated it as a useful future direction (lines 319-321).

R2. the conclusion section does not draw conclusive consideration about the data obtained but repeats the discussion and results 

We appreciate your input on this. The conclusion has been reformulated to focus on what is novel and define the primary conclusions, as well as how this directs next steps and future research.

R2. the discussion and conclusions on the external validity of the results appears lacking 

Thank you for this comment. We have added to the limitations section on the generalizability (external validity) of this study on lines 316-318.

R2. Title: the study design is missing 

We have adapted the title to address this as follows:

A survey of wheelchair service provision education in Canadian occupational therapy programs

R2. It is preferable to not use abbreviations in the abstract, except WHO that is worldwide known to all readers 

Thank you for pointing this out. We have removed “OT” and used occupational therapy instead.

R2. the time necessary to complete the survey could be interesting to evaluate the educators' small number of complete answers 

We were not able to obtain completion time data with our survey platform so it is difficult to make any conclusions about how this impacted responses. After pilot-testing, we indicated to participants that the survey would take 30-40 minutes to complete. As we had a (collective) response from each program and little missing data, we are optimistic that the length of the survey was not a deterrent. We have added in the estimated completion time (based on pilot testing) in the Measurement section on line 127-128.

R2. 29 educators could appear as a high number if the total number of possible educators engageable is not stated 

As per the comment addressed earlier, we believe we obtained >40% of individuals involved in teaching this content area. We added a sentence in the discussion (lines 217-219) to elaborate on the point you have raised here.

R2. It seems unnecessary to report both in minutes and hours the time allocated to the programs; choose one of them 

Thank you, this would simplify things. We have revised the sentence on Line 184 to read “2.5 to 48 hours” and Table 2, as well as in Lines 226 and 238-239.

---

## [Editor Report · Decision Letter 1]

14 Oct 2021

PONE-D-21-27046R1A survey of wheelchair service provision education in Canadian occupational therapy programsPLOS ONE

Dear Dr. Giesbrecht,

Thank you for submitting your manuscript to PLOS ONE. After careful consideration, we feel that it has merit but does not fully meet PLOS ONE’s publication criteria as it currently stands. Therefore, we invite you to submit a revised version of the manuscript that addresses the points raised during the review process.

The paper needs major revisions

We look forward to receiving your revised manuscript.

Kind regards,

Alessandro de Sire, M.D.

Academic Editor

PLOS ONE

Additional Editor Comments (if provided):

The paper needs major revisions
---

## [Author Response · Author response to Decision Letter 1]

14 Dec 2021

Dear Dr. de Sire:

Thank you for your feedback on our manuscript. We have revised our paper to address the concerns as outlined below.

Please revise your Discussion in order to make the paper more intriguing and suitable for publication. I suggest you improve it, citing and discussing recent papers.

We have made revisions to the Discussion section to improve its suitability. In addition to the existing citations, most of which are from 2019-2021, we have introduced four additional citations all of which were published in 2021 and integrated this into the content and application of our findings.

Regards

Ed Giesbrecht

---

## [Editor Report · Decision Letter 2]

19 Dec 2021

Wheelchair service provision education in Canadian occupational therapy programs

PONE-D-21-27046R2

Dear Dr. Giesbrecht,

We’re pleased to inform you that your manuscript has been judged scientifically suitable for publication and will be formally accepted for publication once it meets all outstanding technical requirements.

Kind regards,

Alessandro de Sire, M.D.

Academic Editor

PLOS ONE

---

## [Editor Report · Acceptance letter]

23 Dec 2021

PONE-D-21-27046R2 

Wheelchair service provision education in Canadian occupational therapy programs 

Dear Dr. Giesbrecht:

I'm pleased to inform you that your manuscript has been deemed suitable for publication in PLOS ONE. Congratulations! Your manuscript is now with our production department. 

Kind regards, 

on behalf of

Prof. Alessandro de Sire 

Academic Editor

PLOS ONE